# A New EWMA Control Chart for Monitoring Multinomial Proportions

Shengjin Gan [1,2], Su-Fen Yang [1,*] and Li-Pang Chen [1]

1   Department of Statistics, National Chengchi University, Taipei 116, Taiwan; ganshengjin2001@163.com (S.G.); statphd2016@gmail.com (L.-P.C.)
2   School of Big Data & Artificial Intelligence, Fujian Polytechnic Normal University, Fuqing 350300, China
*   Correspondence: yang@mail2.nccu.tw

**Abstract:** Control charts have been widely used for monitoring process quality in manufacturing and have played an important role in triggering a signal in time when detecting a change in process quality. Many control charts in literature assume that the in-control distribution of the univariate or multivariate process data is continuous. This research develops two exponentially weighted moving average (EWMA) proportion control charts to monitor a process with multinomial proportions under large and small sample sizes, respectively. For a large sample size, the charting statistic depends on the well-known Pearson's chi-square statistic, and the control limit of the EWMA proportion chart is determined by an asymptotical chi-square distribution. For a small sample size, we derive the exact mean and variance of the Pearson's chi-square statistic. Hence, the exact EWMA proportion chart is determined. The proportion chart can also be applied to monitor the distribution-free continuous multivariate process as long as each categorical proportion associated with specification limits of each quality variable is known or estimated. Lastly, we examine simulation studies and real data analysis to conduct the detection performance of the proposed EWMA proportion chart.

**Keywords:** control chart; multinomial distribution; specification limits; Pearson's chi-square statistic

## 1. Introduction

Process control plays a critical role in fostering sustainable practices within industries. It establishes a connection and enables the attainment of secure and efficient process operation and energy systems. Sustainability encompasses the integration of economic, social, and environmental systems, necessitating a well-rounded approach to resource management [1–3]. From the standpoint of process control, several factors contribute to sustainable practices, including the minimization of raw material costs, reduction of product and material scrap/waste expenses, optimization of capital costs, enhancement of process and energy efficiency, mitigation of carbon and water footprints, and maximization of eco-efficiency and process safety. Therefore, process control plays a pivotal role in offering sustainability solutions for developing and implementing efficient technology (refer to Daoutidis et al. [4]). In other words, the practice of sustainability introduces new operational challenges in the development of process control methods. So far, few papers have discussed developing or utilizing control charts to offer sustainability solutions. For example, Anderson et al. [5] applied multivariate control charts to monitor ecological and environmental measurement indices; Morrison [6] used control charts to interpret and monitor environmental data; Gove et al. [7] adopted control charts to catch water supply in south–west Western Australia; Oliveira da Silva et al. [8] constructed control charts to help in stability and reliability of water quality; Shafqat et al. [9] provided triple EWMA mean control chart to monitor and compare Air and Green House Gases Emissions of various countries and identified the critical countries. Control charts serve as effective tools in process control, aiming to enhance the quality and yield of products/parts while reducing scrap/waste of raw materials, minimizing carbon and water footprints, and increasing profits/eco-efficiency and energy efficiency of products.

Among statistical process control tools, control charts are effective tools for monitoring and improving the manufacturing or service process quality. Compared to many process controls with continuous quality variables, less attention has been paid to control charts designed with categorical quality characteristic. The well-known charts for monitoring two-categorical process units are $p, c, np$, and $u$ charts for monitoring nonconforming fraction and defects and for more details refer to Montgomery [10], Reynolds et al. [11,12] and Qiu [13]. However, only considering two categories is not enough to characterize the more general situation of process control. For example, an item can be classified into the three grades of best, better, or good and not just nonconforming and conforming grades. Consequently, the study of process control for categorical data following a multinomial distribution is required to explore carefully.

Up until now, many control charts monitoring multinomial-proportion process are constructed based on Pearson's chi-square statistic, but its variant heavily depends on a large sample size (e.g., Marcucci [14] and Nelson et al. [15]). The asymptotic chi-square distribution of Pearson's chi-square statistic is specifically known for an infinite sample size. When the sample size is small, it is not appropriate to adopt the asymptotic chi-square distribution of Pearson's chi-square statistic to construct the multinomial-proportion control chart because the calculated average run length (ARL) of the asymptotic control charts may seriously deviate from the pre-specified ARL. It thus leads to an over- or under-adjustment of the process.

We note that many papers of multinomial-proportion control charts are designed based on the asymptotic distribution of Pearson's chi-square statistic even when the sample size is small, such as Crosier [16] and Qiu [17]. Moreover, Ryan et al. [18] established the multinomial-proportion CUSUM chart that relies on pre-specified out-of-control multinomial proportions, which consequently leads to worse detection performance compared with multiple one-sided Bernoulli CUSUM charts. Li et al. [19] followed the idea of Qiu [17] to propose an EWMA-type control chart for monitoring the proportions of a multivariate binomial distribution under a large sample size. Huang et al. [20,21] and Lee et al. [22] extended the control chart in Li et al. [19] to monitor the multinomial-proportion process with a large sample size.

From those existing methods, we find that monitoring the multinomial-proportion process with a small sample size has not been discussed. Though the exact distribution of Pearson's chi-square statistic is difficult to know, we may derive its exact mean and variance whether the sample size is small or large. According to the results, we thus provide an exact EWMA-proportion control chart to monitor the multinomial-proportion process. The control limit of the proposed exact control chart can be determined and implemented not only for a small sample size but also for a large sample size and even an individual sample. So far, the literature has not yet discussed the exact EWMA-proportion control chart.

In this study, we have devised a novel, efficient, and accurate method for monitoring and controlling a multinomial-proportion process. The proposed method holds the potential to provide multiple sustainability solutions across industries.

This rest of the paper is organized as follows. Section 2 derives the exact means and variances of Pearson's chi-square statistic under in-control process proportions and studies the properties of Pearson's chi-square statistic. Section 3 constructs the exact and asymptotic EWMA-proportion charts and determine their control limits by satisfying the pre-specified $ARL_0$ and considering small and large sample sizes. Section 4 evaluates and compares the out-of-control proportions' detection performance of the proposed exact and asymptotic EWMA-proportion charts. Section 5 shows how the proposed exact EWMA-proportion chart can be applied to monitor the identify proportions of all categories of a distribution-free continuous multivariate process using a real example of semiconductor data obtained from UCI database. Finally, we offer conclusions of the study.

## 2. Investigation of the Property of Pearson's Chi-Square Statistic for Correlated Quality Variables following a Multinomial Distribution

We first denote $X = (X_1, X_2, \ldots, X_m)$ as the count vector of $m$ categories in $n$ independent trials, where $X_i$ is the count number of the ith category, i = 1, 2,..., $m$. Let $p_0 = (p_{0,1}, p_{0,2}, \ldots, p_{0,m})$ be a vector of the in-control proportion associated with $X = (X_1, X_2, \ldots, X_m)$, where $p_{0,i}$, i = 1, ..., $m$, is the in-control proportion of the i-th category, and $\sum_{i=1}^{m} p_{0,i} = 1$. Next,

$X$ follows a multinomial distribution with probability mass function

$$p(X_1 = x_1, X_2 = x_2, \ldots, X_m = x_m) = \frac{n!}{x_1! x_2! \ldots x_m!} p_{0,1}^{x_1} p_{0,2}^{x_2} \cdots p_{0,m}^{x_m},$$

where $\sum_{i=1}^{m} x_i = n$, and $x_i$ is the realization value of $X_i$ for i = 1, ..., $m$.

To know whether there is a change in the in-control proportion vector $p_0$, a natural idea is to adopt the Pearson's chi-square statistic to make a test. The in-control Pearson's chi-square statistic:

$$\chi^2 = \sum_{i=1}^{m} \frac{(X_i - e_{0,i})^2}{e_{0,i}}, \tag{1}$$

where $e_{0,i} = n p_{0,i}$ is the in control expected number of the ith category.

We now study the in-control distribution of the Pearson's chi-square statistic and derive its exact mean and variance by considering various sample size and in-control proportion vector. When $n$ is large enough, the Pearson's chi-square statistic $\chi^2$ follows an asymptotical chi-square distribution with degree of freedom (df) $m - 1$; that is, $\chi^2 \sim \chi^2(m - 1)$. This is a well-known asymptotical distribution. When $n$ is small, the distribution of Pearson's chi-square statistic does not follow the $\chi^2(m - 1)$ distribution. Hence, it is better to know the distribution of the Pearson's chi-square statistic for a small sample size. However, it is impossible to know the exact distribution of the Pearson's chi-square statistic, but we may derive its exact mean and variance as follows.

First, it is easy to derive the in-control mean of Pearson's chi-square statistics $\chi^2$ given the in-control proportion as follows.

$$E(\chi^2) = \sum_{i=1}^{m} \frac{p_{0,i}(1 - p_{0,i})}{p_{0,i}} = \sum_{i=1}^{m} (1 - p_{0,i})$$
$$= m - 1 \tag{2}$$

As per our best knowledge, the variance of the Pearson's chi-square statistic has not been derived. We derive the in-control exact variance of Pearson's chi-square statistic $\chi^2$ as follows.

$$Var(\chi^2) = \sum_{i=1}^{m} \frac{1}{n p_{0,i}} - \frac{m^2 + 2m - 2}{n} + 2(m - 1) \tag{3}$$

The Appendix A presents the derivation process. From (3), we find the variance value differs along with sample size $n$ given $m$ and $p_0$, that is, the variance value is not fixed for various $n$.

To investigate how the mean and variance change under different $n$ and in-control proportion vectors, without loss of generality, we consider two scenarios of in-control proportion vectors. In practice, the proportions could be all the same or not. It is the reason that we consider the proportion vector with the two scenarios. The two scenarios of in-control proportion vectors, each with four proportions for four categories are as follows.

Scenario (1): The in-control four proportions are the same,
$p_0 = (0.25, 0.25, 0.25, 0.25)$.
Scenario (2): The in-control four proportions are not all the same,
$p_0 = (0.1, 0.1, 0.4, 0.4)$.

Table 1 shows the calculated exact means and variances under different $n$ and two scenarios of in-control proportion vectors. We find the following results in Table 1:

(i)   Under scenario (1), the exact means are all fixed at 3 whether $n$ is small or large. However, the exact variance increases when $n$ increases but converges to 5.999 when $n$ is equal to 6000.

(ii)  Under scenario (2), the exact mean are all fixed at 3 whether $n$ is small or large. However, the exact variance decreases when $n$ increases but converges to 6.0 when $n$ is equal to 6000.

(iii) The exact variance increases or decreases heavily due to the in-control proportion vector. We can see that the change behavior of the exact variance for increasing $n$ is different in scenarios (1) and (2).

The above results present clear evidence and show that the variance of the Pearson's chi-square statistic is not fixed for a small sample size. However, the variance converges to $2(m-1)$ when the sample size is large enough.

**Table 1.** The exact mean and variance of the Pearson's chi-square statistic for various $n$ under scenarios (1) and (2) with in-control proportion vectors.

| $n$ | Scenario (1) | | Scenario (2) | |
|---|---|---|---|---|
| | $E(\chi^2)$ | $Var(\chi^2)$ | $E(\chi^2)$ | $Var(\chi^2)$ |
| 1 | 3.000 | 0.000 | 3.000 | 9.000 |
| 2 | 3.000 | 3.000 | 3.000 | 7.500 |
| 3 | 3.000 | 4.000 | 3.000 | 7.000 |
| 4 | 3.000 | 4.500 | 3.000 | 6.750 |
| 5 | 3.000 | 4.800 | 3.000 | 6.600 |
| 6 | 3.000 | 5.000 | 3.000 | 6.500 |
| 7 | 3.000 | 5.143 | 3.000 | 6.429 |
| 8 | 3.000 | 5.250 | 3.000 | 6.375 |
| 9 | 3.000 | 5.333 | 3.000 | 6.333 |
| 10 | 3.000 | 5.400 | 3.000 | 6.300 |
| 11 | 3.000 | 5.455 | 3.000 | 6.273 |
| 12 | 3.000 | 5.500 | 3.000 | 6.250 |
| 13 | 3.000 | 5.538 | 3.000 | 6.231 |
| 14 | 3.000 | 5.571 | 3.000 | 6.214 |
| 15 | 3.000 | 5.600 | 3.000 | 6.200 |
| 16 | 3.000 | 5.625 | 3.000 | 6.188 |
| 17 | 3.000 | 5.647 | 3.000 | 6.176 |
| 18 | 3.000 | 5.667 | 3.000 | 6.167 |
| 19 | 3.000 | 5.684 | 3.000 | 6.158 |
| 20 | 3.000 | 5.700 | 3.000 | 6.150 |
| 50 | 3.000 | 5.880 | 3.000 | 6.060 |
| 100 | 3.000 | 5.940 | 3.000 | 6.030 |
| 200 | 3.000 | 5.970 | 3.000 | 6.015 |
| 400 | 3.000 | 5.985 | 3.000 | 6.008 |
| 600 | 3.000 | 5.990 | 3.000 | 6.005 |
| 800 | 3.000 | 5.993 | 3.000 | 6.004 |
| 1000 | 3.000 | 5.994 | 3.000 | 6.003 |
| 2000 | 3.000 | 5.997 | 3.000 | 6.002 |
| 4000 | 3.000 | 5.999 | 3.000 | 6.001 |
| 5000 | 3.000 | 5.999 | 3.000 | 6.000 |
| 6000 | 3.000 | 5.999 | 3.000 | 6.000 |

From Table 1, we can construct the exact EWMA-proportion control chart whether $n$ is small or large.

### 3. A Pearson's Chi-Square ($\chi^2$) Statistic-Based EWMA Chart for Monitoring the Multinomial Proportions

In statistical process control, sample size is usually small and not large. When $n$ is not large enough, the distribution of Pearson's chi-square statistic does not follow the well-known $\chi^2(m-1)$ distribution. The resulting variances of the Pearson's chi-square statistic for various $n$ in Section 2 exhibit this situation. Hence, it is not appropriate to adopt the $\chi^2(m-1)$ distribution to construct the EWMA-$\chi^2$ control chart so as to monitor the multinomial-proportion process. The misuse of the EWMA-$\chi^2$ control chart results in worse out-of-control detection performance.

We are able to derive the exact mean and variance of the Pearson's chi-square statistic whether the sample size is small or not in Section 2, although it is impossible to know the distribution of the Pearson's chi-square statistic. Based on (2) and (3), we may construct the exact EWMA-proportion control chart to monitor the changes in proportion vector of the multinomial quality variables for a small sample size. When sample size $n$ is large enough, the in-control Pearson's chi-square statistic is approximately distributed as $\chi^2(m-1)$ distribution with df $m-1$. Thus, the monitoring statistic is independent of the original multinomial distribution and sample size $n$. Hence, we construct the asymptotic EWMA-proportion control chart. The detection performance of the two proposed EWMA-proportion control charts is then compared.

### 3.1. The Exact Multinomial-Proportion Control Chart

With the derived exact mean and variance of the in-control Pearson's chi-square statistic, we may construct an exact EWMA-proportion control chart with the upper control limit (UCL), center line (CL), and lower control limit (LCL) as follows; see (5), for various sample size. In other words, the EWMA-proportion control chart has the control limit depending the value of $n$ given the $m$ categories. Here, we let LCL be zero since the out-of-control proportion vector leads to an increase in the value of the Pearson's chi-square statistic.

We let the EWMA chart with monitoring statistic $EWMA_{\chi_t^2}$ at time t be the weighted average of the Pearson's chi-square statistic $\chi^2$ at time t:

$$EWMA_{\chi_t^2} = \lambda \chi_t^2 + (1-\lambda)EWMA_{\chi_{t-1}^2}, \ t = 1, 2, \ldots, \tag{4}$$

where $\lambda \in (0, 1)$ is a smooth parameter.

The in-control mean and variance of monitoring statistic $EWMA_{\chi_t^2}$ at time t are

$E(EWMA_{\chi_t^2}) = m - 1$, and $Var(EWMA_{\chi_t^2}) = \left( \sum_{i=1}^{m} \frac{1}{np_{0i}} - \frac{m^2+2m-2}{n} + 2(m-1) \right) \lambda (1 - (1-\lambda)^{2t})/(2-\lambda)$, respectively.

We let $EWMA_{\chi_{t=0}^2} = m - 1$.

The control limits of the exact EWMA-proportion control chart are consequently:

$$UCL_t = m - 1 + L_n \sqrt{\left( \sum_{i=1}^{m} \frac{1}{np_{0i}} - \frac{m^2+2m-2}{n} + 2(m-1) \right) \lambda (1 - (1-\lambda)^{2t})/(2-\lambda)},$$
$$CL_t = m - 1,$$
$$LCL_t = 0, \tag{5}$$

where the coefficient $L_n$ should be chosen to satisfy the specified ARL$_0$.

To determine $L_n$ satisfying a specified ARL$_0$, we use the Monte Carlo method and follow Yang et al. [23]. The Monte Carlo procedure using R program language is applied to calculate $L_n$, by satisfying a specified ARL$_0$ (see Appendix B, Algorithm A1).

Based on the Monte Carlo procedure, Table 2 lists the resulting $L_n$ of the exact EWMA-proportion control charts with specified ARL$_0$ = 370.4 for various combinations of setting $n$ and $\lambda$ under the aforementioned two scenarios with in-control proportion vectors. We find that the $L_n$ value increases slowly as $n$ increases and converges to 2.416 or 2.417 when $n$ is equal to 6000 under scenario (1) or (2).

**Table 2.** The coefficient ($L_n$) of UCL with specified ARL$_0$ = 370.4 for various $n$ and two scenarios of in-control proportion vectors.

| $n$ | $L_n$ | |
|---|---|---|
| | **Scenario (1)** | **Scenario (2)** |
| 1 | - | 2.414 |
| 2 | 2.382 | 2.605 |
| 3 | 2.377 | 2.600 |
| 4 | 2.388 | 2.550 |
| 5 | 2.401 | 2.537 |
| 6 | 2.388 | 2.525 |
| 7 | 2.394 | 2.513 |
| 8 | 2.398 | 2.501 |
| 9 | 2.403 | 2.492 |
| 10 | 2.395 | 2.489 |
| 11 | 2.404 | 2.485 |
| 12 | 2.409 | 2.474 |
| 13 | 2.403 | 2.471 |
| 14 | 2.403 | 2.467 |
| 15 | 2.409 | 2.468 |
| 16 | 2.407 | 2.464 |
| 17 | 2.406 | 2.456 |
| 18 | 2.408 | 2.452 |
| 19 | 2.408 | 2.454 |
| 20 | 2.406 | 2.453 |
| 50 | 2.413 | 2.430 |
| 100 | 2.414 | 2.423 |
| 200 | 2.416 | 2.419 |
| 400 | 2.418 | 2.419 |
| 600 | 2.419 | 2.419 |
| 800 | 2.419 | 2.420 |
| 1000 | 2.419 | 2.420 |
| 2000 | 2.418 | 2.419 |
| 4000 | 2.416 | 2.418 |
| 5000 | 2.416 | 2.417 |
| 6000 | 2.416 | 2.417 |

*3.2. The Asymptotic Multinomial-Proportion Control Chart*

When $n$ is large enough, the Pearson's chi-square statistic $\chi^2$ follows an asymptotical chi-square distribution with df $m-1$ for an in-control process, that is, $\chi^2 \sim \chi^2(m-1)$ with mean $m-1$ and variance $2(m-1)$. Thus, the monitoring statistic is independent of the original multinomial distribution and sample size $n$.

Based on the in-control asymptotical chi-square distribution, we may establish an EWMA multinomial-proportion control chart to monitor whether the proportion vector changes or not.

We let the *EWMA* chart with monitoring statistic $EWMA_{\chi_t^2}$ at time $t$ be

$$EWMA_{\chi_t^2} = \lambda \chi_t^2 + (1-\lambda)EWMA_{\chi_{t-1}^2}, \ t = 1, 2, \ldots, \tag{6}$$

where $EWMA_{\chi_0^2} = E(\chi^2) = m-1$, and $\lambda \in (0, 1)$ is a smooth parameter.

The mean and variance of monitoring statistic $EWMA_{\chi_t^2}$ at time $t$ are $E(EWMA_{\chi_t^2}) = m-1$ and $Var(EWMA_{\chi_t^2}) = 2(m-1)\lambda(1-(1-\lambda)^{2t})/(2-\lambda)$, respectively. We may find that the mean and variance of the monitoring statistic $EWMA_{\chi_t^2}$ are independent on $n$.

Hence, the dynamic control limits of the EWMA-$\chi^2$ control chart are constructed as

$$
\begin{aligned}
UCL_t &= m - 1 + L\sqrt{2(m-1)\lambda(1-(1-\lambda)^{2t})/(2-\lambda)}, \\
CL_t &= m - 1, \\
LCL_t &= 0,
\end{aligned}
\tag{7}
$$

where $L$ is a coefficient of UCL and should be chosen to achieve a specified $ARL_0$.

To determine $L$ satisfying a specified $ARL_0$, we refer to the Markov chain method in Lucas and Saccucci [24] or Chandrasekaran et al. [25]. We describe the $ARL_0$ calculation procedure as follows.

Step 1. For a given $L$, at time $t$, the region $(0, UCL_t]$ is partitioned into $k$(e.g., $k = 101$) subsets or state $A_i$, $i = 1, 2, \ldots, k$, where $A_i = (UCL_t(i-1)/k, \ UCL_t(i)/k]$.

Step 2. Denote the transition probability matrix with transition probabilities $p_{i,j}{}^t$, from state $A_i$ to state $A_j$ at time $t$, as $B_t = (p_{i,j}{}^t)_{k \times k}, t \geq 2$, where

$p_{i,j}{}^t = p(\chi^2(m-1) \leq (UCL_t(j)/k - (1-\lambda)UCL_{t-1}(i-0.5)/k)/\lambda) -$
$p(\chi^2(m-1) \leq (UCL_t(j-1)/k - (1-\lambda)UCL_{t-1}(i-0.5)/k)/\lambda)$.

For $t = 1$,

$p_{i,j}{}^1 = p(\chi^2(m-1) \leq (UCL_1(j)/k - (1-\lambda)UCL_1(i-0.5)/k)/\lambda) -$
$p(\chi^2(m-1) \leq (UCL_1(j-1)/k - (1-\lambda)UCL_1(i-0.5)/k)/\lambda)$.

Step 3. $ARL_0(L) = p^T(Q_1 + 2B_1Q_2 + 3B_1B_2Q_3 + \ldots + nB_1B_2B_3 \ldots B_{n-1}Q_n + \ldots)$, where $Q_t = (I_k - B_t)1$, 1 is a column vector of ones, and the initial state probability is $p = (0, \ldots, 1, \ldots, 0)^T$.

To obtain the coefficient of the UCL, $L$, of the asymptotical control chart we next adopt the bisection algorithm. The calculation procedure is described as follows.

Step 1. For a given in-control $ARL_0$, consider an interval $[L_1, L_2]$ of $L$ such that $ARL_0(L_1) < ARL_0 < ARL_0(L_2)$, and a threshold error $\varepsilon > 0$ (e.g., $\varepsilon = 0.5$),

where $ARL_0(L_1)$ and $ARL_0(L_2)$ are computed by the above-mentioned procedure.

Step 2. Let $L_{middle} = (L_1 + L_2)/2$.

Step 3. If $(ARL_0(L_{middle}) - ARL_0)(ARL_0(L_1) - ARL_0) \leq 0$, then $L_2 = L_{middle}$, else $L_1 = L_{middle}$.

Step 4. Repeat step 2 and step 3 until $|ARL_0(L_{middle}) - ARL_0| \leq \varepsilon$.

Hence, $L = L_{middle}$.

Based on the Markov chain method and bisection algorithm described above, the calculated coefficient ($L$) of the UCL with specified $ARL_0 = 370.4$ under scenario (1) or (2) is 2.416. The result is obvious since $L$ is a fixed value and independent of sample size $n$.

### 3.3. Comparison of the Exact and Asymptotic Multinomial-Proportion Control Charts

The resulting $L$ and $L_n$ of the exact and asymptotic EWMA-proportion control charts for the two scenarios show that $L_n$ converges to $L$ (=2.416) when $n$ ($\geq 6000$) is large enough. However, when $n$ is not large enough, estimated $L_n$ and $L$ exhibit obvious difference. This is evidence that it is incorrect to adopt the asymptotic EWMA-proportion control chart to monitor the multinomial proportion vector when $n$ is small or not large enough. Hence, the exact EWMA-proportion control chart is recommended for small and not large enough $n$.

## 4. Detection Performance Measurement of the Proposed Exact and Asymptotic EWMA-Proportion Control Charts

Without loss of generality, to measure the out-of-control detection performance of the proposed exact and asymptotic EWMA-proportion charts, we consider the following two scenarios with six out-of-control proportion vectors for setting $n = 2(1)20$, 50, 100(100), $\lambda = 0.05$, and $ARL_0 = 370$.

Scenario (1) has in-control proportion vector, $p_0 = (0.25, 0.25, 0.25, 0.25)$, and six out-of-control proportion vectors as follows. The six out-of-control proportion vectors:

$p_1 = (0.2, 0.3, 0.25, 0.25)$, $p_2 = (0.1, 0.4, 0.25, 0.25)$, $p_3 = (0.05, 0.45, 0.25, 0.25)$, $p_4 = (0.2, 0.2, 0.35, 0.25)$, $p_5 = (0.1, 0.1, 0.55, 0.25)$, and $p_6 = (0.05, 0.05, 0.65, 0.25)$.

Scenario (2) with in-control proportion vector, $p_0 = (0.1, 0.1, 0.4, 0.4)$, and six out-of-control proportion vectors run as follows. The six out-of-control proportion vectors: $p_1 = (0.15, 0.05, 0.4, 0.4)$, $p_2 = (0.2, 0, 0.4, 0.4)$, $p_3 = (0.25, 0.25, 0.1, 0.4)$, $p_4 = (0.2, 0.2, 0.35, 0.25)$, $p_5 = (0.15, 0.15, 0.3, 0.4)$, and $p_6 = (0.25, 0.25, 0.25, 0.25)$.

### 4.1. Detection Performance of the Proposed Exact EWMA-Proportion Chart

Applying the calculated control limit coefficient, $L_n$, of the proposed exact chart and the given scenarios (1) and (2) with the six out-of-control proportion vectors and sample size, we can calculate out-of-control average run length (ARL$_1$). The Monte Carlo procedure is also applied to calculate ARL$_1$ using R program language, see Appendix C (Algorithm A2). A smaller ARL$_1$ indicates better detection performance of a control chart. ARL$_1$ is always a popular detection performance index in the study of statistical process control.

The resulting Tables 3 and 4 illustrate the calculated ARL$_1$ (first row) and SDRL (standard deviation of run length; second row) of the proposed exact chart for various $n$ and scenarios (1) and (2), respectively. We find the following results in Tables 3 and 4:

(i)    For detecting any out-of-control proportion vector, ARL$_1$ decreases when $n$ increases;
(ii)   The larger the difference is between $p_0$ and $p_i$, the smaller is ARL$_1$ under each $n$. The result is reasonable.

**Table 3.** ARLs of the proposed exact control chart for various $n$ under scenario (1) with the six out-of-control proportion vectors.

| $n$ | $p_0$ | $p_1$ | $p_2$ | $p_3$ | $p_4$ | $p_5$ | $p_6$ |
|---|---|---|---|---|---|---|---|
| 2 | 369.956 | 321.682 | 121.808 | 65.69 | 243.704 | 32.476 | 13.582 |
|   | 402.099 | 351.861 | 130.346 | 69.036 | 264.746 | 32.604 | 12.771 |
| 3 | 372.065 | 287.588 | 69.136 | 32.504 | 183.376 | 14.306 | 5.923 |
|   | 416.056 | 323.047 | 75.999 | 34.156 | 205.704 | 15.077 | 5.942 |
| 4 | 369.232 | 261.716 | 47.220 | 21.347 | 144.940 | 9.817 | 4.451 |
|   | 393.303 | 278.589 | 47.005 | 19.678 | 153.794 | 8.761 | 3.444 |
| 5 | 370.177 | 238.209 | 32.446 | 14.187 | 114.307 | 6.370 | 2.813 |
|   | 405.620 | 263.725 | 33.244 | 13.570 | 125.545 | 6.160 | 2.369 |
| 6 | 368.793 | 218.664 | 25.131 | 11.102 | 95.834 | 5.307 | 2.577 |
|   | 394.082 | 232.241 | 23.899 | 9.574 | 100.353 | 4.421 | 1.693 |
| 7 | 374.458 | 203.780 | 20.065 | 8.840 | 81.281 | 4.339 | 2.127 |
|   | 398.754 | 217.250 | 18.688 | 7.366 | 84.604 | 3.463 | 1.325 |
| 8 | 369.532 | 185.235 | 16.036 | 6.974 | 67.638 | 3.475 | 1.737 |
|   | 399.416 | 197.368 | 14.924 | 5.832 | 70.737 | 2.815 | 1.051 |
| 9 | 367.247 | 170.07 | 13.245 | 5.749 | 57.690 | 2.899 | 1.487 |
|   | 395.453 | 184.802 | 12.332 | 4.824 | 60.603 | 2.343 | 0.846 |
| 10 | 370.275 | 158.746 | 11.551 | 5.181 | 50.98 | 2.762 | 1.509 |
|    | 396.203 | 167.584 | 10.170 | 3.947 | 52.264 | 1.965 | 0.754 |
| 11 | 370.450 | 146.869 | 9.862 | 4.438 | 44.622 | 2.359 | 1.350 |
|    | 400.534 | 157.557 | 8.811 | 3.391 | 45.979 | 1.715 | 0.635 |
| 12 | 368.108 | 135.948 | 8.451 | 3.764 | 39.605 | 2.106 | 1.215 |
|    | 398.165 | 146.166 | 7.626 | 2.968 | 41.012 | 1.503 | 0.504 |
| 13 | 370.740 | 127.254 | 7.674 | 3.482 | 35.619 | 1.973 | 1.195 |
|    | 398.013 | 134.882 | 6.678 | 2.524 | 36.202 | 1.331 | 0.461 |
| 14 | 369.888 | 119.230 | 6.936 | 3.178 | 32.176 | 1.887 | 1.170 |
|    | 396.682 | 125.792 | 5.874 | 2.246 | 32.313 | 1.183 | 0.418 |
| 15 | 371.409 | 110.564 | 6.162 | 2.785 | 29.037 | 1.697 | 1.110 |
|    | 399.734 | 117.402 | 5.318 | 2.025 | 29.353 | 1.058 | 0.341 |
| 16 | 368.316 | 103.902 | 5.658 | 2.643 | 26.366 | 1.619 | 1.086 |
|    | 396.150 | 110.434 | 4.771 | 1.791 | 26.366 | 0.957 | 0.300 |
| 17 | 372.261 | 97.635 | 5.250 | 2.476 | 24.342 | 1.557 | 1.074 |
|    | 398.352 | 102.595 | 4.308 | 1.609 | 24.132 | 0.875 | 0.274 |

**Table 3.** *Cont.*

| $n$ | $p_0$ | $p_1$ | $p_2$ | $p_3$ | $p_4$ | $p_5$ | $p_6$ |
|---|---|---|---|---|---|---|---|
| 18 | 368.650 | 92.060 | 4.764 | 2.225 | 22.313 | 1.458 | 1.050 |
|    | 397.644 | 97.515 | 3.962 | 1.466 | 22.202 | 0.801 | 0.225 |
| 19 | 369.787 | 86.608 | 4.394 | 2.102 | 20.668 | 1.402 | 1.035 |
|    | 396.360 | 91.298 | 3.594 | 1.345 | 20.551 | 0.726 | 0.189 |
| 20 | 368.262 | 81.618 | 4.127 | 2.004 | 19.156 | 1.359 | 1.030 |
|    | 395.554 | 85.676 | 3.323 | 1.236 | 18.807 | 0.675 | 0.173 |
| 50 | 370.723 | 24.540 | 1.476 | 1.045 | 5.338 | 1.008 | 1.000 |
|    | 398.263 | 24.130 | 0.778 | 0.211 | 4.713 | 0.675 | 0.001 |
| 100 | 370.097 | 9.079 | 1.041 | 1.000 | 2.309 | 1.000 | 1.000 |
|     | 398.439 | 8.360 | 0.203 | 0.009 | 1.678 | 0.002 | 0.000 |
| 200 | 371.126 | 3.564 | 1.000 | 1.000 | 1.286 | 1.000 | 1.000 |
|     | 400.019 | 2.916 | 0.011 | 0.000 | 0.587 | 0.000 | 0.000 |
| 400 | 369.493 | 1.692 | 1.000 | 1.000 | 1.021 | 1.000 | 1.000 |
|     | 398.541 | 1.028 | 0.000 | 0.000 | 0.143 | 0.000 | 0.000 |
| 600 | 370.632 | 1.256 | 1.000 | 1.000 | 1.001 | 1.000 | 1.000 |
|     | 398.363 | 0.542 | 0.000 | 0.000 | 0.033 | 0.000 | 0.000 |
| 800 | 369.187 | 1.101 | 1.000 | 1.000 | 1.000 | 1.000 | 1.000 |
|     | 397.229 | 0.324 | 0.000 | 0.000 | 0.007 | 0.000 | 0.000 |
| 1000 | 369.751 | 1.038 | 1.000 | 1.000 | 1.000 | 1.000 | 1.000 |
|      | 398.334 | 0.196 | 0.000 | 0.000 | 0.001 | 0.000 | 0.000 |
| 2000 | 369.708 | 1.000 | 1.000 | 1.000 | 1.000 | 1.000 | 1.000 |
|      | 398.510 | 0.013 | 0.000 | 0.000 | 0.000 | 0.000 | 0.000 |
| 4000 | 369.557 | 1.000 | 1.000 | 1.000 | 1.000 | 1.000 | 1.000 |
|      | 397.351 | 0.000 | 0.000 | 0.000 | 0.000 | 0.000 | 0.000 |
| 5000 | 369.657 | 1.000 | 1.000 | 1.000 | 1.000 | 1.000 | 1.000 |
|      | 398.279 | 0.000 | 0.000 | 0.000 | 0.000 | 0.000 | 0.000 |
| 6000 | 369.736 | 1.000 | 1.000 | 1.000 | 1.000 | 1.000 | 1.000 |
|      | 398.101 | 0.000 | 0.000 | 0.000 | 0.000 | 0.000 | 0.000 |

**Table 4.** ARLs of the proposed exact control chart for various $n$ under scenario (2) with the six out-of-control proportion vectors.

| $n$ | $p_0$ | $p_1$ | $p_2$ | $p_3$ | $p_4$ | $p_5$ | $p_6$ |
|---|---|---|---|---|---|---|---|
| 1 | 369.314 | 371.081 | 370.828 | 9.320 | 17.190 | 45.580 | 9.318 |
|   | 395.079 | 394.476 | 394.501 | 7.951 | 15.914 | 45.433 | 7.973 |
| 2 | 368.283 | 258.404 | 123.075 | 7.802 | 15.158 | 42.878 | 8.120 |
|   | 400.411 | 283.917 | 138.227 | 6.934 | 14.770 | 44.518 | 7.384 |
| 3 | 369.013 | 207.565 | 74.424 | 4.972 | 11.054 | 34.678 | 5.396 |
|   | 405.564 | 229.870 | 83.969 | 4.754 | 11.299 | 36.799 | 5.359 |
| 4 | 368.840 | 173.702 | 51.568 | 4.441 | 9.838 | 31.085 | 4.930 |
|   | 390.956 | 185.024 | 54.552 | 3.391 | 9.003 | 30.668 | 4.078 |
| 5 | 370.999 | 144.832 | 36.937 | 3.570 | 8.096 | 26.724 | 3.966 |
|   | 395.305 | 157.049 | 38.928 | 2.746 | 7.597 | 26.895 | 3.395 |
| 6 | 370.222 | 123.071 | 27.592 | 2.904 | 6.842 | 23.593 | 3.302 |
|   | 398.943 | 133.663 | 28.795 | 2.217 | 6.532 | 23.916 | 2.841 |
| 7 | 368.671 | 107.071 | 21.611 | 2.494 | 6.081 | 21.262 | 2.970 |
|   | 398.112 | 114.893 | 22.220 | 1.823 | 5.613 | 21.481 | 2.394 |
| 8 | 370.126 | 93.134 | 17.970 | 2.167 | 5.363 | 19.289 | 2.592 |
|   | 395.952 | 99.214 | 17.581 | 1.546 | 4.940 | 19.300 | 2.081 |
| 9 | 370.868 | 81.428 | 14.823 | 2.029 | 4.915 | 17.743 | 2.446 |
|   | 396.084 | 86.310 | 14.296 | 1.318 | 4.388 | 17.596 | 1.829 |
| 10 | 369.120 | 71.317 | 12.402 | 1.789 | 4.354 | 16.071 | 2.139 |
|    | 398.684 | 76.376 | 11.947 | 1.151 | 3.959 | 16.203 | 1.630 |
| 11 | 370.757 | 63.001 | 10.537 | 1.671 | 4.013 | 14.954 | 2.026 |
|    | 398.200 | 67.485 | 10.107 | 1.004 | 3.569 | 14.947 | 1.454 |
| 12 | 368.926 | 57.180 | 9.521 | 1.595 | 3.802 | 14.066 | 1.960 |
|    | 396.388 | 59.868 | 8.605 | 0.889 | 3.222 | 13.791 | 1.306 |

**Table 4.** *Cont.*

| $n$ | $p_0$ | $p_1$ | $p_2$ | $p_3$ | $p_4$ | $p_5$ | $p_6$ |
|---|---|---|---|---|---|---|---|
| 13 | 371.755 | 51.611 | 8.408 | 1.449 | 3.475 | 12.980 | 1.782 |
|    | 398.458 | 53.654 | 7.491 | 0.792 | 2.966 | 12.832 | 1.190 |
| 14 | 369.361 | 46.467 | 7.471 | 1.406 | 3.292 | 12.146 | 1.741 |
|    | 398.027 | 48.400 | 6.571 | 0.715 | 2.725 | 11.953 | 1.096 |
| 15 | 366.476 | 42.014 | 6.654 | 1.331 | 3.002 | 11.312 | 1.599 |
|    | 398.999 | 43.662 | 5.823 | 0.641 | 2.526 | 11.217 | 0.998 |
| 16 | 369.623 | 38.371 | 5.875 | 1.268 | 2.852 | 10.702 | 1.536 |
|    | 398.93 | 39.606 | 1.197 | 0.57 | 2.342 | 10.512 | 0.915 |
| 17 | 372.149 | 35.721 | 5.585 | 1.249 | 2.783 | 10.282 | 1.537 |
|    | 397.024 | 36.112 | 4.611 | 0.531 | 2.171 | 9.860 | 0.862 |
| 18 | 369.494 | 32.851 | 5.151 | 1.215 | 2.634 | 9.769 | 1.461 |
|    | 397.07 | 33.070 | 4.163 | 0.486 | 2.03 | 9.296 | 0.794 |
| 19 | 369.044 | 30.160 | 4.714 | 1.185 | 2.441 | 9.156 | 1.369 |
|    | 398.317 | 30.550 | 3.802 | 0.442 | 1.907 | 8.822 | 0.726 |
| 20 | 369.159 | 27.988 | 4.392 | 1.159 | 2.365 | 8.657 | 1.365 |
|    | 399.616 | 28.106 | 3.473 | 0.410 | 1.797 | 8.356 | 0.690 |
| 50 | 370.314 | 7.236 | 1.420 | 1.000 | 1.242 | 3.407 | 1.019 |
|    | 397.494 | 6.396 | 0.618 | 0.025 | 0.532 | 2.825 | 0.136 |
| 100 | 369.737 | 2.819 | 1.000 | 1.000 | 1.018 | 1.757 | 1.000 |
|     | 398.007 | 2.120 | 0.000 | 0.000 | 0.135 | 1.119 | 0.007 |
| 200 | 369.376 | 1.405 | 1.000 | 1.000 | 1.000 | 1.141 | 1.000 |
|     | 397.284 | 0.709 | 0.000 | 0.000 | 0.007 | 0.391 | 0.000 |
| 400 | 370.64 | 1.031 | 1.000 | 1.000 | 1.000 | 1.005 | 1.000 |
|     | 399.136 | 0.170 | 0.000 | 0.000 | 0.000 | 0.069 | 0.000 |
| 600 | 370.225 | 1.002 | 1.000 | 1.000 | 1.000 | 1.000 | 1.000 |
|     | 398.276 | 0.041 | 0.000 | 0.000 | 0.000 | 0.009 | 0.000 |
| 800 | 370.060 | 1.000 | 1.000 | 1.000 | 1.000 | 1.000 | 1.000 |
|     | 397.990 | 0.008 | 0.000 | 0.000 | 0.000 | 0.001 | 0.000 |
| 1000 | 369.657 | 1.000 | 1.000 | 1.000 | 1.000 | 1.000 | 1.000 |
|      | 398.683 | 0.001 | 0.000 | 0.000 | 0.000 | 0.000 | 0.000 |
| 2000 | 370.317 | 1.000 | 1.000 | 1.000 | 1.000 | 1.000 | 1.000 |
|      | 398.111 | 0.000 | 0.000 | 0.000 | 0.000 | 0.000 | 0.000 |
| 4000 | 370.794 | 1.000 | 1.000 | 1.000 | 1.000 | 1.000 | 1.000 |
|      | 399.123 | 0.000 | 0.000 | 0.000 | 0.000 | 0.000 | 0.000 |
| 5000 | 370.790 | 1.000 | 1.000 | 1.000 | 1.000 | 1.000 | 1.000 |
|      | 399.038 | 0.000 | 0.000 | 0.000 | 0.000 | 0.000 | 0.000 |
| 6000 | 369.862 | 1.000 | 1.000 | 1.000 | 1.000 | 1.000 | 1.000 |
|      | 398.246 | 0.000 | 0.000 | 0.000 | 0.000 | 0.000 | 0.000 |

*4.2. Detection Performance of the Asymptotic EWMA-Proportion Chart*

Applying the calculated control limit coefficient, *L*, of the asymptotic chart and the given scenarios (1) and (2) with the six out-of-control proportion vectors, we can calculate $ARL_1$.

The resulting Table 5 (scenario (1)) and Table 6 (scenario (2)) illustrate the calculated $ARL_1$ (first row) and SDRL (second row) of the asymptotic chart, respectively.

We find the following results in Tables 5 and 6:

(i) Most $ARL_0$s are far away from the specified 370.4 for small *n*. In Table 5, we find many $ARL_0$s are larger than the specified 370.4 for *n* < 400, and some $ARL_1$s are larger than the specified 370.4 for very small *n*. However, in Table 6, we find all $ARL_0$s are smaller than the specified 370.4 for *n* < 6000. These results indicate that the proposed asymptotic control chart is not in-control robust, it becomes ARL biased, and its detection performance is worse for small *n*.

(ii) When *n* is large ($n \geq 400$ for scenario (1) or *n* = 6000 for scenario (2)), the calculated $ARL_0$ close to the specified $ARL_0$, and $ARL_1$ decreases when *n* increases for detecting any out-of-control proportion vector.



(iii) The larger the difference is between $p_0$ and $p_i$, i = 1, 2, ..., 6, the smaller is $ARL_1$ under each $n$.

**Table 5.** ARLs of the asymptotic control chart under various $n$ for scenario (1) with the six out-of-control proportion vectors.

| $n$ | $p_0$ | $p_1$ | $p_2$ | $p_3$ | $p_4$ | $p_5$ | $p_6$ |
|---|---|---|---|---|---|---|---|
| 2 | 3880.926 | 3123.472 | 720.986 | 280.329 | 2074.137 | 100.033 | 32.574 |
|  | 3896.139 | 3131.111 | 713.365 | 267.982 | 2077.971 | 87.278 | 23.585 |
| 3 | 1078.071 | 791.313 | 135.773 | 54.859 | 449.865 | 21.522 | 8.127 |
|  | 1157.757 | 852.399 | 143.038 | 54.858 | 486.158 | 20.860 | 7.673 |
| 4 | 757.384 | 509.243 | 69.903 | 29.123 | 255.223 | 12.387 | 5.275 |
|  | 789.150 | 530.552 | 67.986 | 25.865 | 264.734 | 10.735 | 4.127 |
| 5 | 648.207 | 398.79 | 44.919 | 18.887 | 178.058 | 8.516 | 3.906 |
|  | 671.590 | 412.093 | 41.702 | 15.778 | 181.867 | 6.820 | 2.517 |
| 6 | 569.374 | 321.301 | 30.593 | 12.860 | 129.408 | 5.840 | 2.674 |
|  | 600.160 | 338.397 | 28.619 | 10.987 | 134.551 | 4.960 | 1.853 |
| 7 | 535.804 | 277.828 | 23.219 | 9.835 | 102.369 | 4.649 | 2.184 |
|  | 565.679 | 292.373 | 21.278 | 8.174 | 105.892 | 3.783 | 1.425 |
| 8 | 506.336 | 241.435 | 18.239 | 7.768 | 82.654 | 3.753 | 1.818 |
|  | 538.152 | 255.351 | 16.578 | 6.409 | 85.335 | 3.033 | 1.155 |
| 9 | 483.561 | 212.767 | 14.599 | 6.212 | 68.121 | 3.058 | 1.524 |
|  | 518.434 | 227.899 | 13.408 | 5.205 | 71.033 | 2.507 | 0.909 |
| 10 | 476.051 | 194.730 | 12.641 | 5.506 | 59.056 | 2.837 | 1.515 |
|  | 503.278 | 204.614 | 11.060 | 4.240 | 59.678 | 2.081 | 0.774 |
| 11 | 458.735 | 173.615 | 10.581 | 4.643 | 50.003 | 2.415 | 1.356 |
|  | 490.911 | 184.745 | 9.367 | 3.601 | 51.157 | 1.800 | 0.653 |
| 12 | 455.017 | 160.708 | 9.410 | 4.172 | 44.605 | 2.298 | 1.322 |
|  | 481.168 | 168.485 | 8.035 | 3.048 | 44.578 | 1.549 | 0.577 |
| 13 | 446.672 | 146.102 | 8.163 | 3.641 | 38.955 | 2.015 | 1.200 |
|  | 476.889 | 154.694 | 7.040 | 2.673 | 39.251 | 1.383 | 0.475 |
| 14 | 439.888 | 134.735 | 7.318 | 3.300 | 34.911 | 1.919 | 1.173 |
|  | 468.259 | 141.612 | 6.176 | 2.341 | 34.699 | 1.230 | 0.427 |
| 15 | 437.203 | 125.143 | 6.589 | 3.032 | 31.407 | 1.775 | 1.134 |
|  | 465.765 | 131.462 | 5.493 | 2.066 | 31.184 | 1.100 | 0.372 |
| 16 | 428.399 | 115.217 | 5.884 | 2.715 | 28.267 | 1.636 | 1.086 |
|  | 458.844 | 121.453 | 4.944 | 1.867 | 28.076 | 0.989 | 0.302 |
| 17 | 425.681 | 107.603 | 5.423 | 2.523 | 25.919 | 1.573 | 1.073 |
|  | 454.903 | 112.808 | 4.465 | 1.674 | 25.447 | 0.902 | 0.274 |
| 18 | 420.922 | 100.071 | 4.913 | 2.287 | 23.522 | 1.465 | 1.050 |
|  | 451.455 | 105.644 | 4.088 | 1.532 | 23.301 | 0.815 | 0.228 |
| 19 | 417.849 | 93.837 | 4.547 | 2.148 | 21.729 | 1.411 | 1.036 |
|  | 448.075 | 98.522 | 3.733 | 1.394 | 21.368 | 0.745 | 0.192 |
| 20 | 416.766 | 88.216 | 4.277 | 2.062 | 20.240 | 1.385 | 1.035 |
|  | 445.050 | 92.002 | 3.407 | 1.270 | 19.673 | 0.692 | 0.187 |
| 50 | 386.868 | 25.082 | 1.480 | 1.044 | 5.391 | 1.008 | 1.000 |
|  | 415.975 | 24.631 | 0.785 | 0.21 | 4.773 | 0.090 | 0.000 |
| 100 | 378.202 | 9.145 | 9.082 | 1.000 | 2.319 | 1.000 | 1.000 |
|  | 406.259 | 8.405 | 0.204 | 0.009 | 1.688 | 0.002 | 0.000 |
| 200 | 374.087 | 3.575 | 1.000 | 1.000 | 1.288 | 1.000 | 1.000 |
|  | 403.003 | 2.921 | 0.011 | 0.000 | 0.590 | 0.000 | 0.000 |
| 400 | 370.638 | 1.692 | 1.000 | 1.000 | 1.020 | 1.000 | 1.000 |
|  | 399.267 | 1.028 | 0.000 | 0.000 | 0.143 | 0.000 | 0.000 |
| 600 | 369.798 | 1.256 | 1.000 | 1.000 | 1.001 | 1.000 | 1.000 |
|  | 398.157 | 0.543 | 0.000 | 0.000 | 0.032 | 0.000 | 0.000 |
| 800 | 369.017 | 1.100 | 1.000 | 1.000 | 1.000 | 1.000 | 1.000 |
|  | 397.659 | 0.323 | 0.000 | 0.000 | 0.005 | 0.000 | 0.000 |
| 1000 | 368.672 | 1.038 | 1.000 | 1.000 | 1.000 | 1.000 | 1.000 |
|  | 397.161 | 0.197 | 0.000 | 0.000 | 0.002 | 0.000 | 0.000 |
| 2000 | 369.183 | 1.000 | 1.000 | 1.000 | 1.000 | 1.000 | 1.000 |
|  | 398.185 | 0.013 | 0.000 | 0.000 | 0.000 | 0.000 | 0.000 |
| 4000 | 369.313 | 1.000 | 1.000 | 1.000 | 1.000 | 1.000 | 1.000 |
|  | 398.385 | 0.000 | 0.000 | 0.000 | 0.000 | 0.000 | 0.000 |
| 5000 | 369.596 | 1.000 | 1.000 | 1.000 | 1.000 | 1.000 | 1.000 |
|  | 398.369 | 0.000 | 0.000 | 0.000 | 0.000 | 0.000 | 0.000 |
| 6000 | 369.646 | 1.000 | 1.000 | 1.000 | 1.000 | 1.000 | 1.000 |
|  | 397.875 | 0.000 | 0.000 | 0.000 | 0.000 | 0.000 | 0.000 |

**Table 6.** ARLs of the asymptotic control chart under various $n$ for scenario (2) with the six out-of-control proportion vectors.

| $n$ | $p_0$ | $p_1$ | $p_2$ | $p_3$ | $p_4$ | $p_5$ | $p_6$ |
|---|---|---|---|---|---|---|---|
| 1 | 149.100 | 149.131 | 149.435 | 5.099 | 9.434 | 23.891 | 5.091 |
|   | 190.427 | 190.656 | 190.444 | 6.226 | 11.788 | 30.444 | 6.220 |
| 2 | 211.107 | 156.108 | 81.979 | 6.891 | 12.582 | 31.619 | 7.071 |
|   | 232.441 | 174.418 | 94.030 | 5.926 | 12.043 | 32.925 | 6.270 |
| 3 | 234.377 | 141.543 | 56.129 | 4.239 | 9.132 | 26.670 | 4.632 |
|   | 261.884 | 160.014 | 64.268 | 4.098 | 9.570 | 28.990 | 4.644 |
| 4 | 254.595 | 128.980 | 42.294 | 3.612 | 8.095 | 24.825 | 4.000 |
|   | 278.088 | 140.884 | 45.288 | 3.110 | 8.012 | 25.974 | 3.723 |
| 5 | 270.693 | 114.659 | 31.555 | 3.292 | 7.366 | 23.010 | 3.731 |
|   | 292.512 | 124.793 | 33.353 | 2.500 | 6.881 | 23.390 | 3.122 |
| 6 | 278.487 | 100.133 | 24.204 | 2.654 | 6.237 | 20.532 | 3.071 |
|   | 305.263 | 110.100 | 25.650 | 2.071 | 6.021 | 21.291 | 2.669 |
| 7 | 287.245 | 88.690 | 19.511 | 2.287 | 5.416 | 18.594 | 2.658 |
|   | 315.190 | 97.624 | 20.162 | 1.712 | 5.256 | 19.448 | 2.267 |
| 8 | 297.024 | 80.086 | 16.506 | 2.091 | 5.043 | 17.515 | 2.494 |
|   | 320.759 | 85.897 | 16.214 | 1.454 | 4.642 | 17.787 | 1.970 |
| 9 | 300.812 | 70.928 | 13.705 | 1.919 | 4.657 | 16.204 | 2.369 |
|   | 326.830 | 76.427 | 13.386 | 1.251 | 4.157 | 16.357 | 1.746 |
| 10 | 306.108 | 63.493 | 11.661 | 1.724 | 4.157 | 14.883 | 2.087 |
|    | 331.928 | 68.176 | 11.222 | 1.097 | 3.778 | 15.099 | 1.564 |
| 11 | 309.943 | 56.698 | 9.940 | 1.580 | 3.788 | 13.764 | 1.934 |
|    | 337.242 | 60.932 | 9.547 | 0.959 | 3.422 | 14.016 | 1.400 |
| 12 | 316.717 | 52.133 | 9.015 | 1.539 | 3.694 | 13.238 | 1.936 |
|    | 342.484 | 55.010 | 8.221 | 0.860 | 3.120 | 13.089 | 1.271 |
| 13 | 320.280 | 47.283 | 7.963 | 1.435 | 3.361 | 12.291 | 1.753 |
|    | 346.034 | 49.674 | 7.166 | 0.762 | 2.858 | 12.203 | 1.151 |
| 14 | 321.785 | 42.931 | 7.119 | 1.360 | 3.138 | 11.508 | 1.672 |
|    | 348.787 | 44.946 | 6.303 | 0.683 | 2.637 | 11.437 | 1.055 |
| 15 | 324.025 | 39.232 | 6.411 | 1.324 | 2.937 | 10.800 | 1.583 |
|    | 351.660 | 40.889 | 5.595 | 0.623 | 2.449 | 10.737 | 0.971 |
| 16 | 326.148 | 35.968 | 5.705 | 1.262 | 2.775 | 10.223 | 1.510 |
|    | 353.893 | 37.359 | 5.013 | 0.559 | 2.274 | 10.121 | 0.890 |
| 17 | 329.612 | 33.574 | 5.438 | 1.232 | 2.665 | 9.756 | 1.474 |
|    | 356.022 | 34.347 | 4.462 | 0.514 | 2.118 | 9.515 | 0.830 |
| 18 | 331.238 | 31.008 | 4.978 | 1.189 | 2.541 | 9.284 | 1.432 |
|    | 357.644 | 31.556 | 4.048 | 0.463 | 1.986 | 9.023 | 0.774 |
| 19 | 331.958 | 28.646 | 4.585 | 1.165 | 2.400 | 8.792 | 1.360 |
|    | 359.795 | 29.015 | 3.687 | 0.426 | 1.866 | 8.556 | 0.712 |
| 20 | 333.886 | 26.651 | 4.261 | 1.147 | 2.318 | 8.365 | 1.350 |
|    | 361.667 | 26.966 | 3.367 | 0.395 | 1.751 | 8.096 | 0.675 |
| 50 | 355.057 | 7.161 | 1.417 | 1.001 | 1.241 | 3.38 | 1.019 |
|    | 381.753 | 6.34 | 0.611 | 0.025 | 0.529 | 2.797 | 0.137 |
| 100 | 362.178 | 2.801 | 1.000 | 1.000 | 1.018 | 1.751 | 1.000 |
|     | 391.087 | 2.107 | 0.000 | 0.000 | 0.134 | 1.113 | 0.007 |
| 200 | 366.135 | 1.404 | 1.000 | 1.000 | 1.000 | 1.140 | 1.000 |
|     | 393.971 | 0.708 | 0.000 | 0.000 | 0.007 | 0.390 | 0.000 |
| 400 | 367.412 | 1.031 | 1.000 | 1.000 | 1.000 | 1.005 | 1.000 |
|     | 396.169 | 0.177 | 0.000 | 0.000 | 0.000 | 0.000 | 0.000 |
| 600 | 367.196 | 1.002 | 1.000 | 1.000 | 1.000 | 1.000 | 1.000 |
|     | 396.301 | 0.042 | 0.000 | 0.000 | 0.000 | 0.009 | 0.000 |
| 800 | 367.608 | 1.000 | 1.000 | 1.000 | 1.000 | 1.000 | 1.000 |
|     | 396.326 | 0.008 | 0.000 | 0.000 | 0.000 | 0.001 | 0.000 |
| 1000 | 367.333 | 1.000 | 1.000 | 1.000 | 1.000 | 1.000 | 1.000 |
|      | 395.985 | 0.000 | 0.000 | 0.000 | 0.000 | 0.000 | 0.000 |
| 2000 | 367.691 | 1.000 | 1.000 | 1.000 | 1.000 | 1.000 | 1.000 |
|      | 396.363 | 0.000 | 0.000 | 0.000 | 0.000 | 0.000 | 0.000 |
| 4000 | 368.637 | 1.000 | 1.000 | 1.000 | 1.000 | 1.000 | 1.000 |
|      | 397.286 | 0.000 | 0.000 | 0.000 | 0.000 | 0.000 | 0.000 |
| 5000 | 368.955 | 1.000 | 1.000 | 1.000 | 1.000 | 1.000 | 1.000 |
|      | 397.586 | 0.000 | 0.000 | 0.000 | 0.000 | 0.000 | 0.000 |
| 6000 | 370.236 | 1.000 | 1.000 | 1.000 | 1.000 | 1.000 | 1.000 |
|      | 399.095 | 0.000 | 0.000 | 0.000 | 0.000 | 0.000 | 0.000 |

All those phenomena indicate the asymptotic control chart should be adopted in process control by taking $n \geq 400$ or 6000 in scenario (1) or (2) for the correcting control

process; otherwise, the detection performance of the asymptotic control chart would be worse and result in an incorrect process adjustment.

Compared with the resulting Tables 3–6, we find that the two charts do have almost the same in-control and out-of-control process control performances for $n \geq 6000$. However, the exact EWMA-proportion chart offers correct results compared to the asymptotic control chart, especially for small $n$. Hence, the proposed exact EWMA-proportion chart is recommended whether the sample size is small or not.

## 5. Monitoring Under-Specification Proportions of a Continuous Multivariate Process Using the Proposed EWMA-Proportion Chart and Its Application

The proposed exact EWMA-proportion chart can not only be applied to monitor the proportion vector of a multinomial process but also the proportion vector of multiple categories in a distribution-free or an unknown distributed continuous multivariate process.

In this section, we provide an example to describe how to apply our proposed exact chart to monitor the proportion vector of four categories in a distribution-free or an unknown distributed continuous bivariate process. We adopt a semiconductor manufacturing data-set that can be found in a data depository maintained by the University of California, Irvine (McCann and Johnston [26]). The data-set spans from July 2008 to October 2008 and contains 591 continuous quality variables. Each variable has 1567 observations, including 1463 in-control observations and 104 out-of-control observations.

To demonstrate the detection performance of the proposed exact chart, we select 2 of the 591 continuous correlated quality variables, $X = (X3, X12)^T$. Based on the respective specifications of X3 and X12, they can be classified into four categories. The four categories: (1) X3 and X12 are all under specifications, (2) X3 is under specification, but X12 is not, (3) X3 and X12 are all out of specifications, and (4) X3 is out of specification, but X12 is under specification. By examining the 1463 in-control population observations, we classify their categories and obtain the proportion vector of the four categories as $p_0 = (0.4, 0.08, 0.07, 0.45)$. For the 104 out-of-control population observations, the proportion vector of the four categories is $p_1 = (0.00, 0.00, 0.2167, 0.7833)$. To demonstrate the detection performance of the proposed exact chart, we take the first 100 in-control observations and the first 60 out-of-control observations, respectively. We let the sample size be five, then there are 20 in-control samples and 12 out-of-control samples. To monitor the process proportion vector, we construct the exact control chart applying the aforementioned method.

From (5), we know that the control limit of the proposed exact control chart is variable when sampling time changes. Hence, for each sampling time $t$, we list $UCL_t$, the number of observations in each category ($n_{ij}$), the in-control statistic value ($\chi_t^2$), and charting statistic value ($EWMA_{\chi_t^2}$) for the 20 in-control subgroup data. The results are illustrated in Table 7. We then plot the in-control $EWMA_{\chi_t^2}$ values in the constructed exact control chart; see Figure 1. We find all $EWMA_{\chi_t^2}$ values fall within $UCL_t$ demonstrating that the first 20 samples are all from the population with the in-control proportion vector. Furthermore, we calculate $n_{ij}$, the out-of-control statistic value ($\chi_t^2$), and charting statistic value ($EWMA_{\chi_t^2}$) using the 12 out-of-control subgroup data. The results appear in Table 8. We display the out-of-control $EWMA_{\chi_t^2}$ values in the constructed exact control chart in Figure 2. We find that the first $EWMA_{\chi_t^2}$ value falls outside of $UCL_t$, and ten out of the twelve $EWMA_{\chi_t^2}$ values create signals. It demonstrates that the proposed exact control chart performs well in detecting the out-of-control proportion vector.

**Table 7.** The in-control statistics and UCL of the exact control chart.

| Number $t$ | $n_{11}$ | $n_{12}$ | $n_{21}$ | $n_{22}$ | $\chi_t^2$ | $EWMA_{\chi_t^2}$ | $UCL_t$ |
|---|---|---|---|---|---|---|---|
| 1 | 4 | 0 | 0 | 1 | 3.084 | 3.004 | 3.363 |
| 2 | 3 | 0 | 0 | 2 | 1.146 | 2.911 | 3.500 |
| 3 | 4 | 0 | 0 | 1 | 3.084 | 2.92 | 3.598 |
| 4 | 2 | 2 | 0 | 1 | 7.37 | 3.142 | 3.674 |
| 5 | 1 | 2 | 0 | 2 | 7.337 | 3.352 | 3.735 |
| 6 | 2 | 0 | 0 | 3 | 1.091 | 3.239 | 3.787 |
| 7 | 3 | 0 | 0 | 2 | 1.146 | 3.134 | 3.831 |
| 8 | 1 | 1 | 1 | 2 | 2.694 | 3.112 | 3.869 |
| 9 | 1 | 0 | 1 | 3 | 2.519 | 3.083 | 3.901 |
| 10 | 0 | 2 | 0 | 3 | 9.186 | 3.388 | 3.930 |
| 11 | 4 | 0 | 0 | 1 | 3.084 | 3.373 | 3.955 |
| 12 | 1 | 1 | 1 | 2 | 2.694 | 3.339 | 3.977 |
| 13 | 2 | 0 | 1 | 2 | 1.622 | 3.253 | 3.999 |
| 14 | 1 | 0 | 0 | 4 | 2.918 | 3.236 | 4.017 |
| 15 | 5 | 0 | 0 | 0 | 6.905 | 3.42 | 4.032 |
| 16 | 2 | 0 | 0 | 3 | 1.091 | 3.303 | 4.046 |
| 17 | 1 | 0 | 1 | 3 | 2.519 | 3.264 | 4.058 |
| 18 | 3 | 0 | 1 | 1 | 2.608 | 3.231 | 4.069 |
| 19 | 2 | 0 | 1 | 2 | 1.622 | 3.151 | 4.078 |
| 20 | 0 | 0 | 0 | 5 | 6.628 | 3.325 | 4.087 |

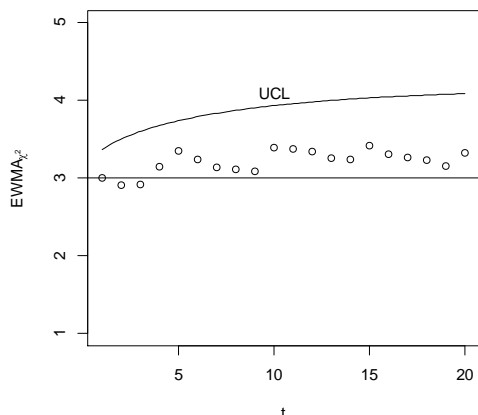

**Figure 1.** The in-control charting statistics on the exact EWMA-proportion control chart.

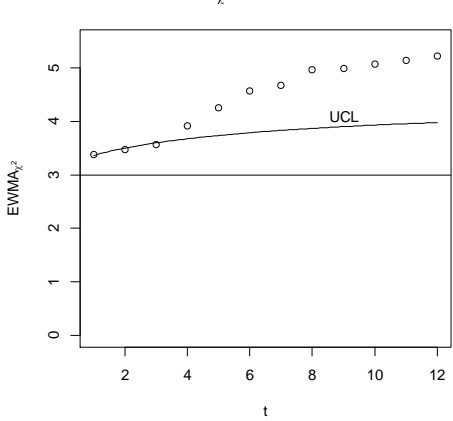

**Figure 2.** The out-of-control charting statistics on the exact EWMA-proportion control chart.

**Table 8.** The out-of-control statistics of the exact EWMA control chart.

| Sampling Time $t$ | $n_{11}$ | $n_{12}$ | $n_{21}$ | $n_{22}$ | $\chi_t^2$ | $EWMA_{\chi_t^2}$ |
|---|---|---|---|---|---|---|
| 1 | 0 | 0 | 2 | 3 | 10.615 | 3.381 |
| 2 | 0 | 0 | 1 | 4 | 5.299 | 3.477 |
| 3 | 0 | 0 | 1 | 4 | 5.299 | 3.568 |
| 4 | 0 | 0 | 2 | 3 | 10.615 | 3.92 |
| 5 | 0 | 0 | 2 | 3 | 10.615 | 4.255 |
| 6 | 0 | 0 | 2 | 3 | 10.615 | 4.573 |
| 7 | 0 | 0 | 0 | 5 | 6.628 | 4.676 |
| 8 | 0 | 0 | 2 | 3 | 10.615 | 4.973 |
| 9 | 0 | 0 | 1 | 4 | 5.299 | 4.989 |
| 10 | 0 | 0 | 0 | 5 | 6.628 | 5.071 |
| 11 | 0 | 0 | 0 | 5 | 6.628 | 5.149 |
| 12 | 0 | 0 | 0 | 5 | 6.628 | 5.223 |

## 6. Conclusions

This paper develops the exact and asymptotic EWMA-proportion control charts to monitor the multinomial-proportions process. Based on the derived in-control exact mean and variance of the chi-square statistic, we calculate the control limits of the exact EWMA-proportion control chart for various small and large sample sizes using the Monte Carlo method. Based on the asymptotic chi-square distribution with df $m - 1$, we calculate the control limits of the asymptotic EWMA-proportion control chart for a large enough sample size using the Markov chain method.

From numerical analyses, we find that control limits (5) and (7) with the same preset in-control ARL and out-of-control detection ability are nearly the same when the sample size is large enough, e.g., n ≥ 6000 under scenarios (1) and (2). For small or moderate sample size, the exact EWMA-proportion control chart is in-control robust, but the asymptotic control chart's in-control ARL is more or less than the preset $ALR_0$ = 370.4. The misuse of the asymptotic control chart results in worse out-of-control detection performance. Thus, we strongly suggest to adopt the proposed exact control chart to monitor a multinomial-proportions process. Moreover, the proposed exact EWMA proportion chart can be adopted to monitor the change in proportions of categories of a distribution-free or unknown continuous distributed multivariate process. A numerical example utilizing semiconductor manufacturing data was discussed to illustrate the application of the proposed exact EWMA proportion chart. The illustration of real data example shows good detection performance of the proposed chart.

In this study, we have developed a novel, efficient, and exact EWMA-proportion control chart specifically designed for monitoring a multinomial-proportion process. Unlike existing literature, which focuses on control charts for multinomial proportions with large or infinite sample sizes, our proposed method is tailored for small and medium sample sizes. Our exact EWMA-proportion control chart offers significant potential for providing sustainable solutions across various industries. We recommend applying this method not only for monitoring multinomial proportions in a multinomial process but also for distribution-free or unknown continuous distributed multivariate processes. By utilizing the proposed exact EWMA-proportion control chart, organizations can effectively monitor and control their processes, enabling them to identify and address deviations or shifts in the multinomial proportions. This approach holds promise for enhancing quality assurance, process optimization, and overall operational performance in diverse industrial settings.

**Author Contributions:** Conceptualization, S.-F.Y. and L.-P.C.; methodology, S.-F.Y. and S.G.; software, S.G.; validation, S.-F.Y.; formal analysis, S.G.; resources, S.-F.Y. and L.-P.C.; data curation, S.-F.Y. and S.G.; writing—original draft preparation, S.-F.Y. and S.G.; writing—review and editing, S.-F.Y.; visualization, S.-F.Y. and S.G.; supervision, S.-F.Y. and L.-P.C.; funding acquisition, S.-F.Y. and S.G. All authors have read and agreed to the published version of the manuscript.

**Funding:** The work was funded by Fujian Polytechnic Normal University (No.HX2022147), China, and Natural Science Foundation of Fujian Province (No.2021J011235),China, and National Science and Technology Council (NSTC 110-2118-M-004-001-MY2), Taiwan.

**Institutional Review Board Statement:** Not applicable.

**Informed Consent Statement:** Not applicable.

**Data Availability Statement:** The data presented in this study are available on request from the corresponding author.

**Acknowledgments:** This study received complete support from Department of Statistics, National Chengchi University, Taiwan, National Science and Technology Council, Taiwan, and School of Big Data & Artificial Intelligence, Fujian Polytechnic Normal University, Fuqing, China.

**Conflicts of Interest:** The authors declare no conflict of interest.

## Appendix A

$X = (X_1, X_2, \ldots, X_m)^T$ is a multinomial distribution associated with size $n$ and probability vector $p_0 = (p_{0,1}, p_{0,2}, \ldots, p_{0,m})$.Thus $X$'s probability density function (pdf) is

$$p(X_1 = x_1, X_2 = x_2, \ldots, X_m = x_m) = \frac{n!}{x_1! x_2! \ldots x_m!} p_{0,1}^{x_1} p_{0,2}^{x_2} \ldots p_{0,m}^{x_m}$$

where $\sum_{i=1}^{m} x_i = n$, $\sum_{i=1}^{m} p_{0,i} = 1$. The marginal pdf of $X_i$, $i = 1, 2, \ldots, m$ is

$$p(X_i = x_i) = \frac{n!}{x_i!(n-x_i)!} p_{0,i}^{x_i}(1 - p_{0,i})^{n-x_i}$$

We then have $E(X_i) = np_{0,i}$, $Var(X_i) = np_{0,i}(1 - p_{0,i})$. Hence, we get:

$$p(X_j = x_j | X_i = x_i) = p(X_j = x_j, X_i = x_i)/p(X_i = x_i)$$
$$= \frac{(n!/x_j!x_i!(n-x_i-x_j)!)p_{0,i}^{x_i}p_{0,j}^{x_j}(1-p_{0,i}-p_{0,j})^{n-x_i-x_j}}{(n!/x_i!(n-x_i)!)p_{0,i}^{x_i}(1-p_{0,i})^{n-x_i}}$$
$$= \frac{(n-x_i)!}{x_j!(n-x_i-x_j)!} \left(\frac{p_{0,j}}{1-p_{0,i}}\right)^{x_j} \left(1 - \frac{p_{0,j}}{1-p_{0,i}}\right)^{n-x_i-x_j}.$$

We immediately see that $X_j | X_i = x_i$ follows a binomial$(n - x_i, \frac{p_{0,j}}{1-p_{0,i}})$ distribution. Now, the following assertion (a) holds.

(a)   $E(X_i - np_{0,i})^4 = np_{0,i}(1 - p_{0,i})(1 + 3p_{0,i}^2 - 3p_{0,i}) + 3n^2 p_{0,i}^2 (1 - p_i)^2 - 3np_{0,i}^2(1 - p_{0,i})^2.$

Proof: suppose that $X_{i1}, X_{i2}, \ldots, X_{in}$ are i.i.d Bernoulli$(p_{0,i})$ and then $X_i = \sum_{j=1}^{n} X_{ij} \sim$ binomial$(n, p_{0,i})$,

$$E(X_i - np_{0,i})^4 = E\left(\sum_{j=1}^{n} (X_{ij} - p_{0,i})\right)^4$$
$$= E\left(\sum_{j_1}\sum_{j_2}\sum_{j_3}\sum_{j_4} (X_{ij_1} - p_{0,i})(X_{ij_2} - p_{0,i})(X_{ij_3} - p_{0,i})(X_{ij_4} - p_{0,i})\right)$$
$$= \sum_{j=1}^{n} E(X_{ij} - p_{0,i})^4 + 3\sum_{j_1=1}^{n}\sum_{j_2 \neq j_1} E(X_{ij_1} - p_{0,i})^2 E(X_{ij_2} - p_{0,j})^2$$
$$= n\left[p_{0,i}^4(1 - p_{0,i}) + (1 - p_{0,i})^4 p_{0,i}\right] + 3n(n-1)p_{0,i}^2(1 - p_{0,i})^2.$$

Under a similar discussion to $E(X_i - np_{0,i})^4$, we can obtain

(b)   $E(X_i - np_{0,i})^3 = \sum_{j=1}^{n} E(X_{ij} - p_{0,i})^3 = n[(1 - p_{0,i})^3 p_{0,i} - p_{0,i}^3(1 - p_{0,i})].$

Thus, we have

$$
\sum_{i=1}^{m} \frac{E(X_i - np_{0,i})^4}{n^2 p_{0,i}^2} = \sum_{i=1}^{m} \frac{1}{np_{0,i}} - \frac{4m-6}{n} - \frac{3\sum_{i=1}^{m} p_{0,i}^2}{n} + 3\sum_{i=1}^{m} (1-p_{0,i})^2 - 3\sum_{i=1}^{m} \frac{(1-p_{0,i})^2}{n}
$$

$$
= \sum_{i=1}^{m} \frac{1}{np_{0,i}} - \frac{4m-6}{n} - \frac{3\sum_{i=1}^{m} p_{0,i}^2}{n} + 3m - 6 + 3\sum_{i=1}^{m} p_{0,i}^2 - \frac{3m-6+3\sum_{i=1}^{m} p_{0,i}^2}{n}
$$

$$
= \sum_{i=1}^{m} \frac{1}{np_{0,i}} - \frac{7m-12+6\sum_{i=1}^{m} p_{0,i}^2}{n} + \sum_{i=1}^{m} 3p_{0,i}^2 + 3m - 6.
$$

For $i \neq j$, we get

$$
E(X_i - np_{0,i})^2 (X_j - np_{0,j})^2 = E\left\{ (X_i - np_{0,i})^2 E[(X_j - np_{0,j})^2 | X_i] \right\}
$$
$$
= E\left\{ (X_i - np_{0,i})^2 [(E(X_j|X_i) - np_{0,j})^2 + Var(X_j|X_i)] \right\}
$$
$$
= E\left\{ (X_i - np_{0,i})^2 \left[ \frac{(X_i - np_{0,i})^2 p_{0,j}^2}{(1-p_{0,i})^2} + (n - X_i)\frac{p_{0,j}}{1-p_{0,i}}\left(1 - \frac{p_{0,j}}{1-p_{0,i}}\right) \right] \right\}
$$
$$
= \frac{p_{0,j}^2}{(1-p_{0,i})^2} E(X_i - np_{0,i})^4 - \frac{p_{0,j}}{1-p_{0,i}}\left(1 - \frac{p_{0,j}}{1-p_{0,i}}\right) E(X_i - np_{0,i})^3 + np_{0,j}\left(1 - \frac{p_{0,j}}{1-p_{0,i}}\right) E(X_i - np_{0,i})^2
$$
$$
= \frac{p_{0,j}^2}{(1-p_{0,i})^2}\left[ np_{0,i}(1-p_{0,i})(1 + 3p_{0,i}^2 - 3p_{0,i}) + 3n^2 p_{0,i}^2 (1-p_{0,i})^2 - 3np_{0,i}^2(1-p_{0,i})^2 \right] -
$$
$$
\frac{p_{0,j}}{1-p_{0,i}}\left(1 - \frac{p_{0,j}}{1-p_{0,i}}\right) n[(1-p_{0,i})^3 p_{0,i} - p_{0,i}^3(1-p_{0,i})] + n^2 p_{0,i} p_{0,j}(1-p_{0,i})\left(1 - \frac{p_{0,j}}{1-p_{0,i}}\right).
$$

Next, we have

$$
\sum_{i=1}^{m} \sum_{j \neq i} \frac{E(X_i - np_{0,i})^2 (X_j - np_{0,j})^2}{n^2 p_{0,i} p_{0,j}}
$$
$$
= \sum_{i=1}^{m} \sum_{j \neq i} \frac{p_{0,j}}{n(1-p_{0,i})}\left[ (1 + 3p_{0,i}^2 - 3p_{0,i}) - 3p_{0,i}(1-p_{0,i}) \right] +
$$
$$
\sum_{i=1}^{m} \sum_{j \neq i} 3p_{0,i} p_{0,j} - \sum_{i=1}^{m} \sum_{j \neq i} \frac{1}{n}\left(1 - \frac{p_{0,j}}{1-p_{0,i}}\right)[(1-p_{0,i})^2 - p_{0,i}^2] + \sum_{i=1}^{m} \sum_{j \neq i} (1-p_{0,i})\left(1 - \frac{p_{0,j}}{1-p_{0,i}}\right)
$$
$$
= \sum_{i=1}^{m} \frac{1}{n}\left[ (1 + 3p_{0,i}^2 - 3p_{0,i}) - 3p_{0,i}(1-p_{0,i}) \right] + \sum_{i=1}^{m} 3p_{0,i}(1-p_{0,i}) -
$$
$$
\sum_{i=1}^{m} \frac{1}{n}(m-2)(1-2p_{0,i}) + \sum_{i=1}^{m} (1-p_{0,i})(m-2)
$$
$$
= \frac{m-6+6\sum_{i=1}^{m} p_{0,i}^2}{n} + 3 - \sum_{i=1}^{m} 3p_{0,i}^2 - \frac{1}{n}(m-2)^2 + (m-1)(m-2).
$$

Furthermore, $\sum_{i=1}^{m} \frac{E(X_i - np_{0,i})^2}{np_{0,i}} = \sum_{i=1}^{m} (1-p_{0,i}) = m - 1$.

Hence, we have

$$
Var\left( \sum_{i=1}^{m} \frac{E(X_i - np_{0,i})^2}{np_{0,i}} \right) = \sum_{i=1}^{m} \frac{E(X_i - np_{0,i})^4}{n^2 p_{0,i}^2} + \sum_{i=1}^{m} \sum_{j \neq i} \frac{E(X_i - np_{0,i})^2 (X_j - np_{0,j})^2}{n^2 p_{0,i} p_{0,j}} - \left( \sum_{i=1}^{m} \frac{E(X_i - np_{0,i})^2}{np_{0,i}} \right)^2
$$
$$
= \sum_{i=1}^{m} \frac{1}{np_{0,i}} - \frac{7m-12+6\sum_{i=1}^{m} p_{0,i}^2}{n} + \sum_{i=1}^{m} 3p_{0,i}^2 + 3m - 6 + \frac{m-6+6\sum_{i=1}^{m} p_{0,i}^2}{n} +
$$
$$
3 - \sum_{i=1}^{m} 3p_{0,i}^2 - \frac{1}{n}(m-2)^2 + (m-1)(m-2) - (m-1)^2
$$
$$
= \sum_{i=1}^{m} \frac{1}{np_{0,i}} - \frac{m^2 + 2m - 2}{n} + 2(m-1).
$$

As $n \to \infty$, $Var\left( \sum_{i=1}^{m} \frac{E(X_i - np_{0,i})^2}{np_{0,i}} \right) \to 2(m-1) = Var(\chi^2(m-1))$.

## Appendix B. R Program Language

**Algorithm A1.** The Monte Carlo simulation steps to find $L_n$ of the exact multinomial-proportion control chart in given $ARL_0$

1: For a given in-control, $p_0 = (p_{0,1}, p_{0,2}, \ldots, p_{0,m}), \lambda, n,$ and specified $ARL_0$ (e.g., $ARL_0 \approx 370$).
2: Set $a < L < b$, e.g., $a = 2$ and b = 3 for $ARL_0 \approx 370$.
3: Monte Carlo procedure:
4: For $N$ from 1 to $M$, set $M$ = 1,000,000 and perform the following:
5: Let $EWMA_{\chi_0^2} = m - 1$, and $t = 1$.
6: Simulate $X_t$ from multinomial distribution with $p_0$ and size $n$, and calculate $\chi_t^2$,
7:   if $t = 1$ then
8: $EWMA_{\chi_1^2} = (1 - \lambda)(m - 1) + \lambda \chi_t^2$.
9:   end if
10:   if $t \neq 1$ then
11: $EWMA_{\chi_t^2} = \lambda \chi_t^2 + (1 - \lambda)EWMA_{\chi_{t-1}^2}$.
12:   end if
13:   Given $L$, and calculate $UCL_t$,
14:   if $EWMA_{\chi_t^2} < UCL_t$, then
15:   $t \leftarrow t + 1$. Go to step line 6.
16:   end if
17: if $EWMA_{\chi_t^2} \geq UCL_t$, then
18: take $t_N = t$ as run length, let $N \leftarrow N + 1$ and go to step 5.
19:   end if
20:   end for
21: Calculate $A\hat{R}L_0 = \frac{1}{M} \sum_{N=1}^{M} t_N$, and determine $L_n$ by $|A\hat{R}L_0 - ARL_0| < 0.8$

## Appendix C. R Program Language

**Algorithm A2**. The Monte Carlo simulation steps to calculate $ARL_1$ of the exact multinomial-proportion control chart

1: For a given in $-$ control, $p_0 = (p_{0,1}, p_{0,2}, \ldots, p_{0,m}), \lambda, n,$ and an out $-$ of $-$ control $p_1$ and $L_n$ obtained by **Algorithm A1** above.
2: Monte Carlo procedure:
3: For $N$ from 1 to $M$, set $M$ = 1,000,000 and perform the following:
4: Let $EWMA_{\chi_0^2} = m - 1$, and $t = 1$.
5: Simulate $X_t$ from multinomial distribution with $p_1$ and size $n$, and calculate $\chi_t^2$.
6: if $t = 1$ then
7: $EWMA_{\chi_1^2} = (1 - \lambda)(m - 1) + \lambda \chi_t^2$.
8: end if
9: if $t \neq 1$ then
10: $EWMA_{\chi_t^2} = \lambda \chi_t^2 + (1 - \lambda)EWMA_{\chi_{t-1}^2}$.
11: end if
12: Given $L_n$, and calculate $UCL_t$,
13: if $EWMA_{\chi_t^2} < UCL_t$, then
14: $t \leftarrow t + 1$. Go to step 5.
15: end if
16: if $EWMA_{\chi_t^2} \geq UCL_t$, then
17: take $t_N = t$ as run length, let $N \leftarrow N + 1$ and go to step 4.
18: end if
19: end for
20: Calculate $A\hat{R}L_1 = \frac{1}{M} \sum_{N=1}^{M} t_N$, take it as an estimator of $ARL_1$.

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
