# Peer review of "A New EWMA Control Chart for Monitoring Multinomial Proportions"

_sustainability, doi:10.3390/su151511797_

Round 1
Reviewer 1 Report
Report on the paper “A new EWMA control chart for monitoring multinomial proportions”
by Gan Shengjin, Su-Fen Yang, Li-Pang Chen
The authors of this paper have created two exponentially weighted moving average (EWMA) proportion control charts that can be used to monitor a process with multinomial proportions, regardless of sample size. For large sample sizes, the chart uses the well-known Pearson Chi-square statistic to determine the control limit, which is based on an asymptotical Chi-square distribution. For small sample sizes, the authors have derived the exact mean and variance of the Pearson Chi-square statistic, resulting in an exact EWMA proportion chart that can be used to monitor the distribution-free continuous multivariate process. This can be done as long as each categorical proportion associated with the specification limits of each quality variable is either known or estimated. The authors have also analyzed the detection performance of the proposed EWMA proportion chart through numerical analyses. They have demonstrated the effectiveness of the proposed charts using real data.
The paper is well-written and provides sufficient background information. It includes numerous references to literature works. Although the mathematics used is common in the statistics literature, they appear sound. What makes this work interesting is the practical application of the proposed method in real-life situations. The results obtained will interest academicians, researchers, and practitioners in product manufacturing. They can use this method to improve the quality and yield of products/parts, reduce scrap/waste of raw materials, minimize carbon and water footprints, and increase products' profits/eco-efficiency and energy efficiency. For better understanding, the authors should provide a reference to the Monte Carlo method used and the software used to obtain numerical results. The conclusion section should link the present finds to that in the existing literature.
Minor editing of the English language required
Reviewer 2 Report
This is a manuscript in which a new control chart for categorical variables with multinomial distribution is proposed. The authors present the demonstrations necessary to understand the performance of the chart.
However, it is not possible to see from the text (and application) how it can contribute to the aspect of sustainable solutions. Only a brief text in the introduction and conclusion mention environmental aspects and sustainability, but superficially.
Thus, I suggest to the authors, an application of the proposed graph to data (real or from the literature) that really lead to environmentally sustainable solutions.
It is also important to provide information about the applied programming language. As it is a new chart, this would allow its applicability.
Author Response
Please see the attached file for all responses.

Reviewer 3 Report
The authors have developed a novel approach for efficient and exact EWMA proportion chart for monitoring a multinomial-proportion process. This is quite essential in providing sustainability solutions in various industries. Their approach, formulation and proposed methodology looks good. There are just a few recommendations for the authors which if they explain or incorporate will make the manuscript more effective. They are mentioned below in random order.
1) It will be good if the authors can mention the time complexity for the methods that they have suggested. Along with that, if they can mention the coding platform and system specifications that they have used in running the simulations and semiconductor manufacturing dataset example.
2) Under section 2, on page 3, the authors have considered scenario 1 and 2 with different in-control proportions. It will be helpful if they can give a justification or reasoning of the particular values of the proportions that they have taken and their relevance in real life manufacturing situations.
3) On line 150, I think it will be 2*(m-1) or 2*df instead of 2m, where df represents degrees of freedom. Because the asymptotic distribution variance is 2*df and the df is m-1.
4) Section 4.2 on line 298, it should be Detecting performance instead of Detection performance
5) In line 111, n will be in line. It is currently showing as a super script.
6) In line 75, it will be According to the results instead of According the results.
7) In line 40, it will be control charts instead of control chars.
Rest looks fine. Best wishes to the authors.
Round 2
Reviewer 2 Report
I still think it is important to apply the proposed chart to environmental data rather than manufacturing industry data.
The authors did not answer which programming language was applied to construct the charts and perform the simulations. Python? R? Other language?
Line 94 - Please correct ARL0 - ARL0 .
